# Influence of Surface Corona Discharge Process on Functional and Antioxidant Properties of Bio-Active Coating Applied onto PLA Films

**DOI:** 10.3390/antiox12040859

**Published:** 2023-04-01

**Authors:** Ana Božović, Katarina Tomašević, Nasreddine Benbettaieb, Frédéric Debeaufort

**Affiliations:** 1Joint Unit Food Processing and Microbiology, Food and Wine Physico-Chemistry Lab, Université de Bourgogne, L’Institut Agro Dijon, 1 Esplanade Erasme, 21000 Dijon, France; ana.bozovic@agrosupdijon.fr (A.B.); katarina.tomasevic@agrosupdijon.fr (K.T.); nasreddine.benbettaieb@u-bourgogne.fr (N.B.); 2MP2, Université Bourgogne Franche Comté, 32 Av. de L’Observatoire, 25000 Besançon, France; 3Institute of Technology, Université de Bourgogne, 7 Blvd Docteur Petitjean, BP 17867, 21078 Dijon, France

**Keywords:** active biobased films, Polylactide, fish gelatin, natural phenolic acid, oxygen permeability, surface treatment properties, DDPH, reducing and chelating properties

## Abstract

PLA (polylactic acid) is one of the three major biopolymers available on the market for food packaging, which is both bio-based and biodegradable. However, its performance as a barrier to gases remains too weak to be used for most types of food, particularly oxygen-sensitive foods. A surface treatment, such as coating, is a potential route for improving the barrier properties and/or providing bioactive properties such as antioxidants. Gelatin-based coating is a biodegradable and food-contact-friendly solution for improving PLA properties. The initial adhesion of gelatin to the film is successful, both over time and during production, however, the coating often delaminates. Corona processing (cold air plasma) is a new tool that requires low energy and no solvents or chemicals. It has been recently applied to the food industry to modify surface properties and has the potential to significantly improve gelatin crosslinking. The effect of this process on the functional properties of the coating, and the integrity of the incorporated active compounds were investigated. Two coatings have been studied, a control fish gelatin-glycerol, and an active one containing gallic acid (GA) as a natural antioxidant. Three powers of the corona process were applied on wet coatings. In the test conditions, there were no improvements in the gelatin crosslinking, but the corona did not cause any structural changes. However, when the corona and gallic acid were combined, the oxygen permeability was significantly reduced, while free radical scavenging, reduction, and chelating properties remained unaffected or slightly improved.

## 1. Introduction

A growing global use of plastics has led to an increased amount of plastic waste in the environment, which causes pollution and harm to the ecosystems. Single-use plastic products are used for a short period of time before being thrown away. The impact of plastic waste on the environment and our health is a global problem [1], however, biodegradable plastics could be part of the solution. Bioplastics can be made from renewable resources, such as crops and wood, or from waste streams like the residues of food processing [2]. One such example is polylactic acid (PLA), which is currently one of the most popular thermoplastics, derived from lactic monomers extracted from natural sources like fermented sugarcane or corn [3]. PLA is one of the biodegradable plastics commercially available for packaging, however, its biodegradability is effective only in tailored conditions of temperature and humidity (industrial composting) [4]. PLA characteristics are similar to polypropylene (PP) and polyethylene (PE) [5]. Nevertheless, the problem with this bioplastic film is its weak gas-barrier property, so it cannot be used for products that are sensitive to external factors like oxygen and humidity. The casting technique, which consists of depositing a thin layer of coating on the film, is a simple and proven method to solve this problem, and optimize the permeability of the PLA films [4,6]. Indeed, Svagan et al. [7] applied thin coatings based on an organic-inorganic phase (chitosan and montmorillonite) on the PLA film that decreased the oxygen permeability by about 100 times. More recently, Debeaufort et al. [8] decreased the oxygen permeability by 600 times thanks to fish gelatin based coatings, according to the relative humidity range.

Many polymeric films offer very low surface energy values, and therefore low adhesion properties. For this reason, a range of surface treatments, including chemical, thermal, and electrical methods are used to aid the adhesion of the coating, particularly in case of the PLA film [3,4]. The most interesting of these methods is one based on plasma technology, due to its low environmental impact. Corona discharge plasma technology, i.e., cold air plasma, promotes high surface energy values which improves the adhesion [5,9]. Both UV illumination (generated by the corona discharge) and the free radicals coming from the ionized gas modify the surface properties of the polymer film by etching or adding functional groups, and oxidizing the polymer chains of PLA, making the surface more hydrophilic [10]. However, these could also induce the denaturation of natural biomacromolecules such as deoxyribonucleic acid (DNA) by modifying its structure and its bioactivity [11], packaging surface decontamination [12], inactivation of enzyme or microbial toxins, or bioactive natural compounds such as antioxidants and antimicrobials [13]. Stepczynska [14] also demonstrates the ability of the corona discharge treatment for the decontamination of the PLA films, whereas Rocca-Smith et al. [15] found that the corona treatment on commercial PLA films favored its hydrolytic degradation.

Coatings can improve the PLA water vapor and oxygen barrier properties [7,8]. Current coatings are mainly low-cost and easily available synthetic polymers with many environmental drawbacks [16]. Incidentally, alternative biopolymer coatings, based on renewable resources, are being developed. Besides being environmentally more favorable, these coatings can also be used as vehicles for the carrier and release of antioxidant and/or antimicrobial compounds. Such active packaging delays the microbial spoilage and increases shelf-life [17]. One biopolymer used to improve the barrier properties, and maintain both the food contact compatibility and the biodegradability of the PLA packaging films, while also being a carrier of active compounds is gelatin [18]. When PLA films are coated with gelatin and the humidity is kept below 80–85%, the permeability of gases (mainly O_2_) can be decreased up to 600 times [8]. The main drawback of this type of coating is its loss of mechanical integrity during industrial processing, which can cause breakage or coating delamination. Therefore, methods which would improve the gelatin crosslinking and the adhesion-like corona discharge process, while maintaining its biodegradability/biocompatibility and the integrity of the incorporated active compounds are sought after. Phenolic acids are well-known natural antioxidants, often recovered from the byproducts of fruit juice processing, or from lignin waste from the paper industry [19]. Some of the phenolic acids are a well-known crosslinker for proteins like tannic acids and its oligomers [20]. Gallic acid (GA) has high antioxidant properties and is able to be released into food without significant limitations or health risks in the concentration range necessary for its expected bioactivity. GA also promotes the gelatin crosslinking [21] and is a good candidate for the development of active coating based on gelatin [22].

Besides modifying the surface properties of the PLA film, as previously mentioned, the corona discharge treatment has been suggested to affect or favor protein crosslinking [23]. Due to their oxidizing effect on amino acids, corona-produced radical species make the side chains of amino acids more reactive, and could potentially initiate gelatin crosslinking [24]. The corona discharge treatment has only recently been introduced to the food industry, despite its long-time use in other industries. Currently, the main concern is the possible effect that the corona-induced free radicals could have on the nutritional quality of the food, or the bioactive components of the packaging.

This study aims to examine whether the efficiency of the corona-produced free radicals affect the crosslinking of gelatin, improving its film barrier properties while maintaining the antioxidant capacity of the PLA-coated active films intended for food contact material.

## 2. Materials and Methods

### 2.1. Materials

The commercial fish gelatin (ref. CO-SP-004, Louis François, Marne-La-Vallée, France) with a Bloom degree = 200, viscosity = 3.5–4.5 mPa.s at 60 °C and at a concentration of 6.67% in water at pH = 5.8, and a pI (isoelectric point) = 4.8–5.3, was used for the coatings. Anhydrous glycerol (98% purity, Fluka-Fisher Scientific SAS, Illkirch, France) was used as a plasticizer for the coating.

The commercial and biodegradable semi-crystalline polylactic acid (PLA) film (Nativia-NTSS-25, Taghleef Industries, San Giorgio di Nogaro, Italy), which the both sides have been previously corona treated during its industrial production, was the base film with a nominal thickness of 25 µm. All the available characteristics of the commercial PLA NTSS-25 films used in this study are available from the manufacturer and are given as Appendix A.

The gallic acid is from Sigma-Aldrich Chimie SARL (Saint-Quentin Fallavier, France), with a minimum purity 98%, molecular weight *M_w_* = 170.12 g·mol^−1^, molar volume *V_M_* = 97.3 cm^3^·mol^−1^, density at 20 °C = 1.749 g·cm^−3^, melting temperature *T^mp^* = 251 °C, boiling temperature *T^bp^* = 501 °C, and the physical-chemical properties data, which has been retrieved from www.ChemSpider.com [accessed on 25 October 2022]), are Log*P* (water octanol partition coefficient) = 0.964, p*Ka* (dissociation constant) = 4.09 in water or aqueous solution, solubility in water = 11.5 g·L^−1^ at 20 °C. The gallic acid was selected as a natural antioxidant added in the gelatin coating to provide bioactive properties to the films.

### 2.2. Coating Suspension and Coated-Film Preparations

A stock suspension of 15% (*w*/*w*) gelatin was prepared in distilled water (500 g) at 50 °C, pH = 5.8–6.0, for each formulation. The temperature of 50 °C was maintained until the coating step. After 30 min of moderate magnetic stirring (about 100 rpm), glycerol (10% of the gelatin weight) was added, and the mixture was further stirred for 10 min. Gelatin-glycerol film-forming suspension was then cast directly on the PLA films (50 cm × 20 cm rectangle pieces). For the samples with the phenolic acid, 5% (to the gelatin weight) of the gallic acid was added after the glycerol was dissolved. The suspension was stirred for an additional 20 min prior to coating.

A thin-layer chromatography applicator (Desaga Brinckmann, DS200/0.3 TLC, Heidelberg, Germany) was used to coat the PLA base films. The PLA films were fixed on a plexiglass plate prior to the casting process using adhesive tapes. The gelatin suspension was then poured into the applicator (set to have a liquid thickness of 250 µm) and then cast at a constant speed of about 20 cm·s^−1^. Six trials of each formulation were casted. After 3 to 5 min at room temperature for setting the gelation of gelatin, different corona-discharge treatments were applied (120 W, 420 W and 870 W) on the wet gelled coatings. A HF Corona Generator CG061P (Arcotec GmbH, Monsheim, Germany) was used for the manual application of the corona discharge treatment at a speed of about 2 cm·s^−1^. The corona equipment characteristics were (i) for the power input: 230 V, 50 Hz, 4.5 A, max 1000 W, and (ii) output electrode: 3–20 kV (adjustable), 25–50 kHz, 200 mm electrode manual roller, and a 460 × 460 mm^2^ counter electrode plate. The corona discharge application on the wet coating principle scheme and photos are in Figure 1 (the video of the application process is available on demand to authors).

The coated PLA films (treated and untreated) were then dried for 24 h at 25 ± 2 °C and 30 ± 5% relative humidity (RH). The coated films and the control films (PLA uncoated) were then conditioned for at least 72 h at 25 °C and 50% RH. The sample coding is presented in Table 1.

### 2.3. Surface Properties Measurements

Goniometry is a method used to determine the physicochemical properties of an interface between two phases (solid/liquid) by measuring the contact angle, and calculating the surface energy (drop method), the surface tension (hanging drop method), and the adhesion energy (drop method and hanging drop) [25,26,27]. These surface characterizations have been considered for the packaging film properties, particularly for the printing or coating properties of the polymers and biopolymers [9,28].

Measurement of the contact angle and surface energy, by the sessile drop method, was done using a Kruss goniometer (DSA30, Kruss GmbH, Villebon, France), and was performed for four liquids covering a wide range of surface tension, including water, glycerol, ethylene-glycol, and diiodomethane. A drop of tested liquid (~3 μL) was automatically placed on the PLA film or coating surface. Both the contact angle and drop volume were measured for three minutes using a camera and the Kruss-Advance operating software (version 2) to estimate the contact angle between the baseline of the drop and the tangent at the drop boundary, as well as the drop spreading rate (kinetics of wetting). Data experimentally acquired were θ, the contact angle, AS, the droplet surface area and V, the droplet volume, as a function of time t. Experiments were performed in a conditioned room at a temperature of 25 ± 2 °C and a relative humidity of 50 ± 1%, and at least 3 replicates were made. In addition, all films were preconditioned in a chamber at the same environmental conditions to avoid interferences due to concurrent moisture exchange at the surface around the droplet. All the measurements of the coated samples were done on the gelatin side only.

The relationship between surface tensions at a three-phase contact point between a solid surface S, a liquid L and its vapor V is described by the following Young’s Equation [29]:(1)$$\gamma^{LV}\times cos(\theta )=\gamma^{SV}–{\gamma^{SL}}_{}$$
where *γ*^LV^, *γ*^SV^ and *γ*^SL^ are the surface tensions (or free energy per unit area) of the liquid–vapor, solid–vapor, and solid–liquid interfaces, and *θ* is the contact angle.

The surface free energy (*γ*^S^) of a solid (expressed in mJ·m^−2^) is closely related to its wettability. *γ*_S_ can be referred to as the excess energy at the surface of a material, compared to that in the bulk [30]. Every system strives for the lowest possible state of free energy. Solids cannot minimize their surface by deformation, however, they can form an interface with a liquid to reduce the free energy, i.e., they can be wetted [31]. High *γ*^S^ corresponds to good wettability, and can be calculated from the contact angle *θ*, the known surface tension, and the polar and dispersive components of the liquids used [30,32].

Theoretically, the contact angle goes from 0° up to 180°, where 0° would represent complete spreading of the liquid on the solid surface. The highest surface tension of a liquid that can completely wet a specific solid surface (i.e., the critical surface tension of the solid material, denoted by *γ*^Crit^), defines the wettability of the solid surface [32]. It is calculated by the intercept of the horizontal line cos *θ* = 1 with the extrapolated straight-line plot cos *θ* versus *γ*^LV^ (known as Zisman plot [27,31]) using different liquids. The value of the contact angle θ with water indicates how hydrophobic the surface is. A large contact angle is representative of a hydrophobic surface, whereas a small contact angle implies a hydrophilic surface.

When a water droplet is deposited on a surface, its volume and contact angle decrease over time due to evaporation and/or absorption. The kinetics of volume change allow us to estimate the absorption rate (expressed in µL·mm^−2^·s^−1^) for hydrophilic surfaces. Indeed, the absorption rate normalized by the surface area of the liquid ingress could be considered as an estimation of the liquid water transfer rate (LWTR) [33]. The surface water absorption rate (SWAR) was thus calculated according to Hambleton et al. [34] using Equation (2):(2)$$SWAR=\frac{{dV}^{abs}}{A^{B}\left(t\right)\times dt}=\frac{dV\left(t\right)-dV^{eva}(t)}{A^{B}\left(t\right)\times dt}=\frac{dV\left(t\right)-[F^{eva}\times dt\times A^{S}\left(t\right)]}{A^{B}\left(t\right)\times dt}$$
where d*V*(t) is the drop volume variation, d*V*^eva^(t) is the evaporated volume variation (µL) and d*V*^abs^(t) is the absorbed volume variation (µL), A^B^(t) is the drop base surface, A^S^(t) is the drop area exposed to air, according to time t, and *F*^eva^ is the evaporation flux calculated according to Equation (3):(3)$$F^{eva}=\frac{V^{eva}\left(t\right)-V^{eva}(t+dt)}{A^{S}\left(t\right)\times dt}=\frac{dV^{eva}(t)}{A^{S}\left(t\right)\times dt}$$

### 2.4. Water Solubility and Water Capacity Absorption of Coatings

Gelatin-based coatings are water sensitive and their behavior in contact with liquid water is an important factor in the development of food contact packaging. Therefore, about 300 mg of coating layer weight was immersed in 20 mL of distilled water at 25 °C for 24 h. Samples were then filtered on filter paper under a vacuum (Buchner) until no more filtrate drop could be recovered. The coating sample retained on the paper filter was weighed just after filtration, and then after 48 h of drying at 60 °C (to prevent thermal degradation and Maillard reactions). The solubility percentage and water absorption capacity were calculated according to the Equations (4) and (5).
(4)$$Solubility(\%)=\frac{m_{initialbeforeimmersion}-m_{retainedafterdrying}}{m_{initialbeforeimmersion}}$$
(5)$$Waterabsorptioncapacity=\frac{m_{retainedbeforedrying}-m_{retainedafterdrying}}{m_{initialbeforeimmersion}}$$
where *m*_initial before immersion_ is the amount of coating before being immersed in the water, *m*_retained before drying_ and *m*_retained after drying_ are the respective weights of the coating retained on the filter paper before and after the 48 h drying at 60 °C. The solubility of the coating was expressed as a percentage of the initial weight of the film, and the water absorption capacity as a gram of water per gram of initial weight of coating (*m*_water_·*m*^−1^_coating_). The amount of recoverable filtrate was also weighed to confirm the capacity of absorption of the gelatin coating.

### 2.5. Water Vapor and Oxygen Permeability Determinations

The gravimetric cup method, described in ISO 2528 [35] and adapted to PLA and other biopolymers by Debeaufort et al. [36], was used to measure the water-vapor transmission rate (WVTR) and the water vapor permeability coefficient (WVP). The experiment was done using the relative humidity gradient of 30–75% at a temperature of 25 ± 1 °C. All film samples were conditioned for at least 72 h at 25 ± 1 °C and at 50% RH before testing. Individual samples were placed with the coated side down in the permeation cells containing NaCl solution (75% RH) and were held in a ventilated chamber (KBF 240 Binder, Tuttlingen, Germany) with controlled temperature and relative humidity (25 °C and 30%). Weighing of the cells was done once per day until linearity was reached. For each type of film, the average value of four thickness measurements was used in the WVP computations (statistical error on the film thickness was considered in WVP uncertainty). The following Equation (6) was used to compute the WVTR (g·m^−2^·s^−1^) from the linear portion of the cell mass variation (weight loss) versus time:(6)$$WVTR=\frac{\Delta m}{A\times \Delta t}$$
where *A* is the area of the film exposed to moisture transfer (9.08 × 10^4^ m^2^) and ∆*m*/∆*t* is the mass loss per unit of time (g·s^−1^). Four replications were done for each sample.

WVP (g·m·m^−2^·s^−1^·Pa^−1^) was determined using the Equation (7):(7)$$WVP=\frac{\Delta m}{A\times \Delta t\times \Delta p}\times e$$
where ∆*p* is the difference in the water vapor partial pressure between the two sides of the film (Pa), and *e* is the thickness of the film (m). The relative humidity value and the saturated water vapor pressure at the measurement temperature (25 ± 1 °C) are used to determine the water vapor partial pressure. Each sample was replicated four times.

The series resistance model was used to calculate the water vapor permeability of the coating “alone”. According to this approach, the sum of the transfer resistances of each layer determines the resistance to the total transfer of water vapor [37,38]. The ratio between thickness and permeability is equivalent to transfer resistance. Knowing this, following Equations (8) and (9) can be derived:(8)$${\left(\frac{e^{coatedPLA}}{{WVP}^{coatedPLA}}\right)}^{measured}={\left(\frac{e^{coating}}{{WVP}^{coatedPLA}}\right)}^{calculated}+{\left(\frac{e^{PLA}}{{WVP}^{PLA}}\right)}^{measured}$$
(9)$${{(WVP}^{coating})}^{calculated}=\left(\frac{e^{coating}}{{{\left(\frac{e^{coatedPLA}}{{WVP}^{coatedPLA}}\right)}^{measured}}_{}-{\left(\frac{e^{PLA}}{{WVP}^{PLA}}\right)}^{measured}}\right)$$

Dealing with the oxygen transfer rate (OTR) and permeability (OP), a Brugger GTT permeameter (Brugger Feinmechanik GmbH, Munich, Germany) was used to determine the oxygen transfer through the coated films, using the manometric (pressure difference) method in accordance with ISO 15105-1 [39]. The film samples were placed in a cell with two compartments: one was blown with oxygen and maintained at a controlled relative humidity, while the other was first subjected to a high vacuum prior to starting the experiment, and then after being airtightly closed, the pressure increase was measured over time. The measuring chamber’s temperature was set at 25 °C.

The software displays the oxygen transmission rate (OTR, cm^3^·m^−2^·s^−1^) graphically as a function of time until reaching the steady state of permeation, at which point the OTR value is recorded. The measurement was performed at a relative humidity of 50%, and at least three repetitions were done for each sample.

To evaluate the oxygen barrier properties of each film formulation, oxygen permeability (OP, cm^3^·m·m^−2^·s^−1^·Pa^−1^) was calculated by multiplying the given OTR with the sample’s thickness and divided by the oxygen pressure differential (corresponding to the absolute pressure difference measured). Furthermore, the OP of the gelatin layer alone was evaluated for each coated sample using the following Equation (10), as done for water vapor permeability:(10)$${{(OP}^{coating})}^{calculated}=\left(\frac{e^{coating}}{{{\left(\frac{e^{coatedPLA}}{{OP}^{coatedPLA}}\right)}^{measured}}_{}-{\left(\frac{e^{PLA}}{{OP}^{PLA}}\right)}^{measured}}\right)$$

### 2.6. Fourier Transform Infra-Red Characterization

Fourier-transform infrared spectroscopy (FTIR) was used to study the molecular interactions between polymers and reactive compounds, and well as changes in the molecular structure of the coating. FTIR spectrophotometer (Spectrum 65; Perkin-Elmer, Haguenau, France), equipped with an ATR attachment with a ZnSe crystal, was used. Measurements were done in the range from 4000 to 600 cm^−1^, at room temperature (25 ± 2 °C), using a resolution of 2 cm^−1^, and 64 scans. The Spectrum Suite software was used to evaluate the measured spectra, and the results were represented as transmittance. Duplicate measurements were done for every sample. Only shifts of peaks for a particular chemical group interacting within the coating were considered, as ATR mode does not allow accurate quantification of peak intensity. The shift in peaks of Amides I and II were checked, as they correspond to the groups more susceptible to react or interact in proteins.

### 2.7. Antioxidant Activity Assessment

The antioxidant activity can be assessed using several tests that measure different parameters, such as the radical species consumed or generated, the compounds resulting from the oxidation reactions, reducing, or chelating powers to metal ions. In this study, the capacity to scavenge a free radical, as well as the capacity to reduce or chelate iron ions, was used for the characterization of the active coatings based on gelatin, containing the gallic acid or not.

#### 2.7.1. Free Radical Scavenging (DPPH) Test

The DPPH (2,2-diphenyl-1-picryl-hydrazyl-hydrate) is a frequently used method for assessing the capacity of natural antioxidants to scavenge free radicals. The method published by Siripatrawan and Harte [40], and adapted for edible films by Benbettaïeb et al. [41], was used to evaluate the activity of antioxidants in the coated samples, based on their capacity to scavenge the DPPH• radical. A piece of 2 × 12 cm^2^ was cut from each sample and placed inside a glass vial containing 10 mL of DPPH solution (50 mg·L^−1^) in ethanol (96% *v*/*v*). A DDPH test was also applied on the coating only, after delamination from PLA, and on pure gelatin powder and pure gallic acid, in the same proportion as in the coating part of coated PLA films. Aluminum foil was placed over the glass vials to shield the solution from the light. Using a UV spectrophotometer (Biochrom WPA Lightwave II UV/Visible spectrophotometer, Cambridge, UK), the absorbance measurement at 515 nm after 1 h and 24 h revealed the extent of the disappearance of the DPPH• reactant. DPPH represents a stable free radical which is nitrogen-based and purple colored. This color can be identified as an absorbance peak at 515 nm. After accepting an electron or hydrogen radical from the antioxidant molecule, the violet color of the reactant changes to a pale yellow. At this point, DPPH turns into a stable diamagnetic molecule, which results in the decreasing of the absorbance value at 515 nm. The antioxidant activity, AA (%), calculated using Equation (11) represents the radical scavenging activity, and is expressed as a percentage (%) of DPPH inhibition.
(11)$$AA\left(\%\right)=\frac{A^{blank}-A^{sample}}{A^{blank}}\times 100$$
where *A*^blank^ is the absorbance of DPPH solution and *A*^sample^ is the absorbance of the DPPH solution with film samples. Three measurements were done for each sample.

#### 2.7.2. Iron Ion Reducing Capacity Test

The reducing power indicates the ability of an antioxidant to reduce a ferricyanide complex to a ferrous form, which shows the ability of this compound to give an electron to the iron ion.

The ability of the antioxidant to reduce a ferricyanide complex to a ferrous form were determined according to the method reported by Mathew and Abraham [42], and adapted by Benbettaieb and al. (2020) [43]. The iron reducing capacity assay is based on the principle that substances, which have reduction potential, reacts with potassium ferricyanide (K_3_Fe^3+^(CN)_6_) to form potassium ferrocyanide (K_4_Fe^2+^(CN)_6_), which then reacts with ferric chloride (FeCl_3_) to form a potassium ferrocyanide-ferric complex (KFe^3+^Fe^2+^(CN)_6_) having Perl’s Prussian blue color and maximum absorption at 700 nm [44]. The reactional mechanism and equations were reported in-depth in a previous work [43]. 10 mg of the coating sample was immersed in a hemolysis tube (5 mL) and mixed with 1.75 mL of distilled water and 1.25 mL of 1% (*w*/*v*) potassium ferricyanide (K_3_Fe (CN)_6_) solution previously prepared in water. The mixtures were maintained at 50 °C for 3 h in a water bath. To stop the reaction, 1.25 mL of trichloroacetic acid (TCA at 10% (*w*/*v*)) was added to the mixture. Finally, 1.25 mL of the mixture solution of each sample was mixed with 1.25 mL of distilled water and 0.25 mL of ferric chloride (FeCl_3_ at 0.1% *w*/*v*). After 10 min of reaction, the absorbance was measured at 700 nm against the blank value. The result was discussed regarding the value of the absorbance. The higher the absorbance is, the higher the reducing ability is. The mean value was calculated from the triplicate analyses. The reducing power ability of the pure fish gelatin and the gallic acid was also assessed. The amount of these compounds (in powder) was calculated to be equal to the theoretical amount existing in the 10 mg of the coating.

#### 2.7.3. Iron Ion Chelating Power Test

The method for measuring the chelation of ferrous iron by an antioxidant was adapted from [45]. The reaction between ferrozine and ferrous iron (Fe^2+^) was yielded to ferrous-ion-ferrozine red color complex having a maximum absorbance at 562 nm. The presence of chelating agents inhibits the formation of this complex and therefore leads to a quantitative decrease of the red color. The measurement of color reduction gives an estimation of the chelating ability of the antioxidant. The higher the absorbance at 562 nm (which is due to the ferrous ion-ferrozine complex), the weaker the ferrous iron chelating power of the antioxidant. 10 mg of coating was immersed in Eppendorf tubes of 2 mL containing 0.1 mL of 2 mM FeCl_2_-4H_2_O and 1.1 mL of distilled water. The mixtures were kept at 25 ± 2 °C for 5 min. The reaction was initiated by the addition of 0.4 mL of ferrozine (3-(2-Pyridyl)-5, 6-diphenyl-1,2,4-triazine-p,p’-disulfonic acid monosodium salt hydrate) solution at 5 mM initially prepared in water. The mixtures were shaken and standing at 25 ± 2 °C for 10 min. The absorbance was then read at 562 nm against a blank (1.5 mL of distilled water + 0.1 mL of 2 mM FeCl_2_-4H_2_O). Control tubes were prepared in the same manner as the sample but without the addition of coating. The chelating power of the pure fish gelatin and the gallic acid was also tested. The amount of these compounds (in powder) was calculated to be equal to the theoretical amount existing in 10 mg of coating.

The percentage of inhibition of ferrozine–Fe^2+^ complex formation was expressed as iron chelating power (ICP, %) and calculated using the Equation (12):(12)$$ICP\left(\%\right)=\frac{A^{control}-A^{sample}}{A^{control}}\times 100$$
where *A*^control^ and *A*^sample^ represents the absorbance of the control and the tested sample tubes, respectively. Measurements were done in triplicate.

### 2.8. Statistical Analyses

Statistical analyses of the data were performed with SPSS 13.0 software (SPSS Inc., Chicago, IL, USA). Using the software, a one-way analysis of variance (ANOVA) was done in order to determine the significant difference through multiple comparisons of means. A Tukey test was applied at a significance level of 95% (*p* < 0.05) for the antioxidant activity test, water, and oxygen permeabilities, and for all surface property parameters, which were carried out in triplicate. The mean values given in tables or graphs with the same letter are not significantly different from each other (*p* > 0.05 ANOVA followed by the Tukey test)

## 3. Results

The coating process and the corona treatment have been conducted according to the protocols described in the materials section. All coatings had similar thicknesses, varying between 20 and 30 µm, which confirms that the manual coating was almost repeatable. However, when the highest power of corona was used (870 W), the PG-870 sample became brownish and shrunken, displaying a surface degradation of the gelatin layer, probably due to a higher temperature exposure. Indeed, Kchaou et al. [46] have displayed that fish gelatin is able to oxidize in the presence of glycerol when exposed to heat, by the way of the Maillard reaction. Therefore, the results of the PG-870 analyses are more affected by this thermal degradation induced by the highest power of the corona discharge process.

### 3.1. Coating Layer Behavior When Exposed to Liquid Water

Water sensitivity of the coating is crucial for applications in contact with highly hydrated foods. Therefore, assessing its capacity to resist liquid water is of great importance. Indeed, it could be linked to crosslinking, or, on the contrary, its capacity to absorb gives an indication for the swelling of the gelatin layer and the release of the active compounds entrapped in the coating. Table 2 displays the results of the water solubility of the coating layers, their capacity to absorb water, and the amount of filtrate recovered after filtration which is related to water absorption capacity.

The coating solubility is greater for the non-corona-treated film. This is not significant because of the very high standard deviation of the non-treated films, but it reveals that the gelatin network is less homogeneous and less stabilized for the PG and PGA. This let us assume that the corona discharge, even if weak, influenced the network structure, trending to reduce the solubility by 30 to 50%. The lowest solubility observed for the PG-870 is mainly attributed to the thermal effect and the Maillard reaction enhancing the crosslinking as displayed by Kchaou et al. [46] or Etxabide et al. [47]. The same trend is confirmed only for the PG-series films for which the corona treatment reduced the water absorption capacity. The addition of gallic acid had an unexpected effect because it doubled the films capacity to absorb water and swell, confirmed by a significant reduction in the volume of the recovered filtrate. Gallic acid and other phenolic acids are known to favor the reticulation of proteins like gelatin, displayed often by a reduction of solubility, but this is not directly linked to the swelling capacity [48,49]. The gelatin-gallic acid coating behaves like a sponge, which means that it is able to retain a high amount of water at the PLA film surface, susceptible to act as a plasticizer. This effect is checked by examining barrier properties of films.

### 3.2. Molecular Interactions Induced by Either Gallic Acid and/or Corona Discharge

Fourier transform infrared spectroscopy (FTIR) was performed on the ATR (Attenuated Total Reflectance) mode on the coated side to observe and identify possible new interactions occurring inside the gelatin network, or between the gelatin and gallic acid. The ATR mode allows for the surface analysis of a sample with weak deepness, including from nanometer to micrometer deepness according to the refractive/reflective indexes of surface samples. Preliminarily FTIR analysis of the PLA (non-coated sample) were done to identify peaks only related to the PLA structure. This allowed us to check if the PLA would interfere with the coating side observation. Indeed, the ATR mode which measures the spectra only in the thin part of the surface did not detect the PLA peaks in the coated films measurement. Therefore, this confirmed that only the coating part, without the PLA-coating interface, was observed. Therefore, chemical structural changes in the coating layer can be inferred from the detection of new peaks and/or of shifts in peaks from the spectra of different samples.

Gelatin-based film spectra are mainly discussed in reference to its protein amide groups. Previous studies assigned the amide A, B, I, II, and III bands of gelatin at wave numbers 3272 cm^−1^, 2924 cm^−1^, 1630 cm^−1^, 1539 cm^−1^, and 1235 cm^−1^ respectively. These peaks are in accordance with our results, and previous studies [8,41], as displayed in Figure 2. Amide A bands overlaps with OH bands of water and makes gelatin interactions difficult to discuss in this region. However, it seems that the OH band is more important for PGA-series films and could be linked to their higher capacity to absorb water. To avoid this interference, the observed spectra were focused from 2000 cm^−1^ to 600 cm^−1^. All formulations (with and without gallic acid) and all corona treatments (0 to 870 W power) showed the same peaks, characteristic of the gelatin spectrum. The absence of new peaks means that no new covalent bonds were formed in the coating with the gallic acid addition, nor after the corona treatment. However, the observed slight shift in peaks is indicative of new low-energy interactions, such as hydrogen or dipole-dipole bonds, similar to the ones already present between gelatin chains. The most sensitive spectral region to the gelatin’s secondary structure is the amide I band (in the range 1700–1600 cm^−1^), which is mainly due to the C–O stretch vibrations of the peptide linkages (~80%) and less due to the C–C stretching [50]. The ν_C–O_ (1640 cm^−1^) is significantly shifted by an 8 cm^−1^ increase when gallic acid was added, and by 4 to 6 cm^−1^ when the corona discharge was applied. These amide I shifts towards higher wavenumbers could be attributed to a change in the secondary structure of gelatin from ⊎-sheet (maximum at 1630–1645 cm^−1^) toward a more random structure [50,51]. Shifts of the amide II band also confirmed this assumption. Indeed, the values of around 1650–1655 cm^−1^ are attributed more to the ⊎-sheet whereas ⊍-helix is centered at 1655 up to 1680 cm^−1^.

As the ATR mode does not quantify bonds based on peak intensity, so we cannot be sure which treatment resulted in more new interactions. Nevertheless, we could state that no degradation of gelatin happened, and no new covalent bonds formed as no new peaks appeared. We could thus suppose that the corona discharge had a very weak effect on the deep structural modification, affecting only the tridimensional organization at the surface. Being well known to generate radicals, corona discharge could also interfere with the antioxidant properties of the film components.

### 3.3. Surface Properties Behaviours

Surface properties influence moisture transfer through the film by promoting or restricting the spreading of liquids on the surface, thereby significantly impacting the shelf-life of packaged foods. As mentioned previously, high *γ*^S^ corresponds to good wettability i.e., good spreading of liquid on the film’s surface. Critical surface free tension (*γ*^Crit^) defines the wettability of a solid by giving a value for the highest surface free tension of a liquid that would completely spread on the solid i.e., liquids having *γ*^L^ lower than the critical will completely spread on the surface. Corona treatments have shown a desirable effect on the surface properties of the coated films. Results are presented in the Table 3.

The water contact angles of the coated films are all increased compared to that of the PLA support films. The NTSS-25 commercial films had been surface corona treated during the industrial process to enable the coating and printing process which increased the hydrophilicity of the PLA surface, as previously demonstrated [52]. All the gelatin-based coating without gallic acid exhibit a non-significantly modified contact angle regardless of the corona power. The value of approximately 81–84° are in accordance with the values obtained by Białopiotrowicz and Jańczuk [53] for gelatin film surfaces at the same concentration. Gallic acid addition, coupled with corona discharge, induced a high hydrophilization of the surface expressed by the contact angle reduction. This behavior could also be seen as very common because a surface treated by corona discharge has a lowered contact angle of water.

As expected, the PLA had the lowest *γ*^S^ due to the absence of a hygroscopic coating. *γ*^S^ values for PG and PG-870 showed high variance between samples. In the case of PG, this was attributed to the swelling of the surface which made the angle calculations difficult. When it comes to PG-870, obvious surface heterogenicity and thermal degradation were resultant from a too-powerful corona treatment, as previously mentioned. The corona discharge power of 120 W and 420 W seemed to promote the lowest *γ*^S^ values. This effect was synergistic with the addition of GA, as the lowest wettability was seen for PGA-120 and PGA-420. This could appear as contradictory to the contact angle behavior, however, the contact angle of water is highly dominated and driven by the polar component of the surface tension of the water.

The critical surface tension of the PLA is much lower than those of the coatings tested, revealing that those liquids having a surface tension ranging from 25–29 mJ·m^−2^, such as common solvents (propanol, dichloromethane, cyclohexane), will easily spread on the PLA. *γ*^Crit^ of the coatings are not significantly affected by the corona treatment, nor by the GA addition. Coatings will be more easily wetted by liquids that have a higher *γ*^L^ (36–42 mJ·m^−2^), like most of the aqueous solutions and food liquids (milk, oils, wine) [54], or polyols often used as plasticizers in edible films and coatings [55].

Even if the PLA is considered as having a low water absorption capacity, its SWAR is not negligible. The surface water absorption rate was by far the highest for the gelatin-coated film without treatment, which is related to the swelling and fast deformation of its surface, leading to inconsistent angle measurements. Corona application or gallic acid addition reduced the absorption rate by at least double, decreasing from 26 × 10^4^ to about 11 × 10^4^ µL·mm^−2^·s^−1^ respectively for PG and all other coatings. Indeed, when the corona treatment and gallic acid addition were combined, they showed a synergistic effect confirming the *γ*^S^ behavior.

### 3.4. Barrier Properties

#### 3.4.1. Water Vapor Transfer Behavior

The measured values of the water vapor permeability (WVP) for the PLA, the PLA/gelatin, and the PLA/gelatin-GA films, both untreated and corona treated, and those calculated from a series resistance model for the coatings alone are presented in Figure 3. Only WVP is discussed and not the WVTR, as WVP normalized the transfer rate by the thickness and partial pressure gradient. The WVP value measured for the PLA is in accordance with the datasheet of the manufacturer, after corrections according to the RH differential were used, as well as in the same order of magnitude to that measured by Shogren [56].

The addition of the coating tended to significantly increase the WVP of the PLA film (from 1.56 × 10^−11^ to 2.35 × 10^−11^ g·m^−1^·s^−1^·Pa^−1^). Indeed, the transfer in the gelatin layer, which is more water-absorptive than the PLA, is favored since the coated side was exposed to the highest humidity. As a result, the coating serves as a reservoir of water at the PLA surface, as displayed from the water absorption capacity determinations, facilitating plasticization and transfer. This can be understood by the fact that coating causes plasticization of the PLA, making it more mobile, which facilitates the diffusion of small molecules. The WVP calculated value of the coating alone are in accordance to those measured by Etxabide et al. [57] for fish gelatin waste films. The WVP of fish gelatin coating is considered as non-effective and very high, due to the water absorption capacity of gelatins which is well-known [58].

Compared with the non-treated PLA/gelatin film, the trend of decreasing WVP can be observed regardless of the corona treatment power used. This is explained by the limitation of the swelling of the coating induced by the corona discharge process and observed during the contact angle kinetic measurements.

The addition of gallic acid decreased WVP due to its crosslinking ability. However, in treated PGA samples, a negative effect of corona can be observed. Potentially, the treatment partly disturbed the interaction between the gallic acid and the gelatin. It could also be said that it could degrade the gallic acid to a certain amount, but this event was not confirmed by the results of the antioxidant properties (discussed later). If there is no crosslinker (gallic acid) added to the coating the corona treatment had a positive effect, because it acts as a crosslinker itself. However, when phenolic acid is incorporated, which is seemingly more efficient than corona, it decreases the efficacy of the added substance. It can be concluded that the corona treatment and gallic acid do not have a synergistic effect on WVP, but rather an antagonistic one.

To check the effect of the side exposed to moisture and the water reservoir phenomena, the WVP was also measured when the PLA side was exposed to the higher RH (Figure 4). In the PLA/gelatin samples, almost in all cases of the corona treatment, the WVP was not significantly affected when the face exposed to higher humidity is PLA. For the lowest and highest treatment power, there is a slight decrease of permeability compared to the non-treated one. This is interesting to know for the application in food packaging. Even if the coated PLA film is used to pack dry food, and the coating is on the side of lower relative humidity, we could have a slight decrease of permeability. This confirms the “reservoir” effect of the gelatin layer when exposed to high RH.

From a water barrier efficacy point of view, the gelatin-based coating did not significantly or positively enhance the moisture barrier efficacy of the coated PLA films. Almost no publication displayed an improvement of the water vapor barrier properties of the PLA films coated with hydrophilic biopolymers, such as proteins or carbohydrates. Only when nanoparticles were incorporated in chitosan, a 20% decrease of the WVP was observed, but was not enough to significantly prolong the foods shelf-life in such packaging [59].

#### 3.4.2. Oxygen Transfer Properties

The presence of oxygen is the most common contributor to the quality and/or safety reduction of the packaged food. Therefore, the oxygen barrier property of the film is a key parameter which justifies its use for packaging applications. As reported by Ledari et al. [60], the increase of the crosslinkages between the gelatin chains due to their interaction with a plasma-induced radical species, decreases the free volume of the gelatin network. Therefore, it can be expected that the penetration of oxygen into the polymer will thereby be reduced. Indeed, oxygen permeability reduction was significantly observed in all coated films.

Figure 5 displays the expected OP decrease when the coatings were applied to the PLA. The OP decrease trend continued when comparing the corona-treated formulations with the untreated ones, except for the PG-870 degraded sample. Furthermore, all formulations containing gallic acid showed a significantly lower OP, up to about 100 times, compared to the ones containing only gelatin. Hydrophilic coatings on the PLA often permit several order of magnitudes reduction of the OP, as displayed by Han et al. [59] for chitosan based coatings applied on the PLA film. Indeed, thin layers of chitosan decrease the OP of coated PLA, compared to neat PLA by almost 100 times. The same conclusions have been given for gluten coating laminated onto the PLA that allowed to drop down the OP by 1000 to 10,000 times according to thicknesses [61].

The calculated OP of the standalone coatings showed that the corona treatment seemed to not improve the O_2_ barrier property of the gelatin standalone coatings, which might suggest that the observed OP decrease might be mainly due to the gelatin coating.

### 3.5. Antioxidant Efficacy of Coated PLA Films

#### 3.5.1. Free Radical Scavenging Activity

The antioxidant activity of the PLA-coated films and of the coatings separated from PLA by delamination, both corona-treated and untreated, as well as the pure gelatin and gallic acid, was evaluated using a method based on the scavenging of the DPPH radical in an ethanol solution. Three experiments were done with different DPPH concentrations (5, 20 and 50 mg L^−1^),film areas (12 or 24 cm^2^), and coating, including gelatin and gallic acid weight corresponding to that of 24 cm^2^ coated films, to optimize the conditions for the observation and display the release kinetics influence.

Values of the antioxidant activity (AA, %) of the PG and the PGA samples were calculated as described in the Methods section. The DPPH concentration of 5 mg L^−1^ was too low and, consequently, the absorbance values were too low to be accurately discussed. When 20 mg L^−1^ of DPPH and 12 cm^2^ of the sample were used, the values of antioxidant activity of the samples with GA were similar after 1 h and 24 h. This indicates that the DPPH completely reacted with the samples during the first hour of the reaction. Therefore, there was not enough DPPH to react for the longer times. In order to monitor the influence of the kinetics of the release, we increased the concentration of DPPH up to 50 mg L^−1^ and the film area to 24 cm^2^. The antioxidant activities for these samples after 1 h and 24 h are presented in Table 4 and confirmed that the antioxidant reaction goes longer than 1 h.

The intrinsic radical scavenging activity of the PG films was the lowest for all samples tested. However, there was a small increase in the antioxidant activity of these samples with time, which can be linked to gelatin hydrolysis. Hydrolysates of gelatin can act as natural antioxidants by scavenging free radicals [46]. When it comes to fish gelatin, the main attributor to the antioxidant activity is the peptide fraction of protein, which can act as an electron or hydrogen donor.

All the PLA films coated with the mixture of gelatin and gallic acid had significantly higher AA values compared to the PLA/gelatin films. This confirms that the antioxidant activity of coated PLA films is coming primarily from the gallic acid. In general, phenolic compounds can stabilize free radicals either by being electron or hydrogen donors, or by delocalizing free electrons in the aromatic ring [18]. When using a high initial concentration of DPPH, after 1 h, there is around 50% of DPPH remaining in the solution. However, after 24 h, the AA increased by about 50%. This suggests a progressive release of the gallic acid from the coating into the DPPH medium over time. The coating allows for a controlled release of the antioxidant.

No significant differences were observed in the antioxidant activity of the PGA-based films between non-treated films and those treated with corona (whatever the power). Although corona did not increase the antioxidant capacity of the samples, it did not decrease it either. These results give an important conclusion: the corona treatment has no negative effect on the free radical scavenging capacity of PLA films coated with the gelatin-gallic acid systems. In other words, if the natural phenolic compounds are incorporated in the coating, they will not be destroyed by the casting and the corona discharge processes and remain able to diffuse into the food.

#### 3.5.2. Iron Reducing and Chelating Powers

The results of both the reducing power and chelating power confirm those of the DPPH and are presented in Table 5. Indeed, the gallic acid has the highest power to reduce or chelate iron ions. Compared to DPPH data, gelatin exhibits a significant reducing power. This property of gelatin to reduce iron was previously demonstrated by several authors [62,63]. Dealing with the chelating power, gelatin also has a very important effect, even if lower than that of gallic acid. However, after repeating the measurements, the higher value of the chelating power of the PG-120 cannot be clearly explained.

The antioxidant properties of the coated films are high enough for packaging oxygen--sensitive foods. Using the corona discharge treatment on undried gelatin-based coatings does not significantly affect or degrade the bioactive properties of the phenolic acid incorporated into the films .

## 4. Conclusions

FTIR spectroscopy showed that corona treatment does not enhance the creation of covalent bonds in the coated PLA films but increases the formation of weaker interactions involved in the crosslinking of gelatin coating, such as hydrogen or dipole-dipole bonding.

Based on the goniometry measurements, it seems that the synergistic action of the addition of gallic acid and the corona process improved the surface properties and reduced the water absorption rate.

The application of corona on gelatin-based coating prior to drying allowed for a lower water absorption rate for all formulations and modified the surface energy of the coating differently, according to the incorporation or exclusion of gallic acid. Water vapor permeability of the PLA is slightly increased due the moisture absorption capacity of gelatin that induced PLA plasticization. The corona treatment allowed to reduce weakly the WVP of PG-coated filmswhereas it slightly increased that of the gelatin-gallic acid-coated samples. This leads to the conclusion that the corona could improve the barrier properties of the films, except for when the phenolic acid is incorporated before the treatment, because then the corona could have a negative impact on the barrier properties. Contrarily to moisture transfers, the oxygen barrier efficacy of the films was significantly improved by both the coating application, the corona discharge treatment, and the gallic acid addition.

An important observation was that the corona discharge did not affect the antioxidant activity of the gallic acid incorporated into the gelatin layer. This suggests that corona could be used on active bio-based packaging that incorporates natural compounds, such as polyphenols, without a negative effect on their antioxidant efficacy.

Since corona is a low-energy, non-toxic treatment already staged in industrial processing, it supports the development of bio-based, sustainable food packaging. Further investigation is needed to better understand the impact and optimize the application of Corona discharge process on gelatin coating in packaging applications.

## Figures and Tables

**Figure 1 antioxidants-12-00859-f001:**
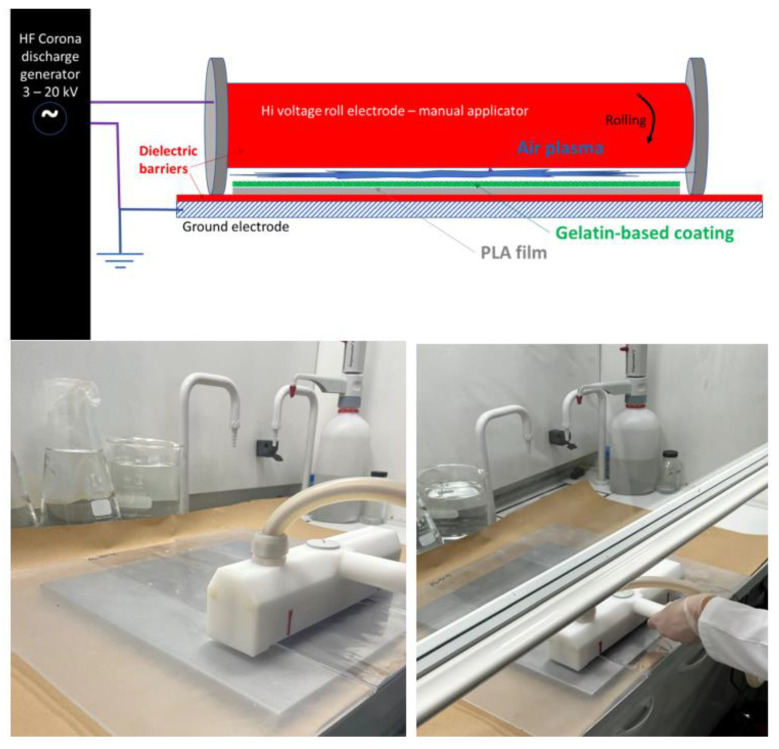
Corona discharge principle and equipment used to apply the Corona air plasma onto the coated PLA with the gelatin-based gel: scheme and picture.

**Figure 2 antioxidants-12-00859-f002:**
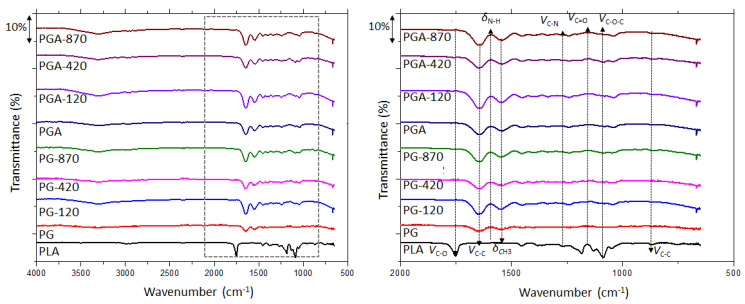
FTIR spectra of PLA and coated films. Left: Zoom of the fingerprint region showing characteristic bond vibrations in 500–2000 cm^−1^. PLA, poly(lactic) acid; PG, PLA-gelatin; PGA, PLA-gelatin-gallic acid.

**Figure 3 antioxidants-12-00859-f003:**
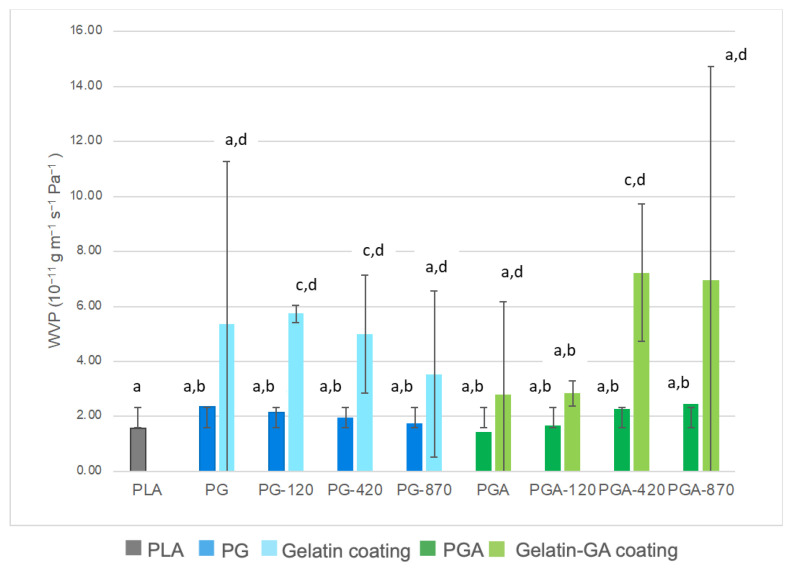
Measured WVP of the PLA and the coated PLA samples (gelatin side exposed to RH 75%, PLA side to RH 30%) and calculated WVP for the coating alone (^a,b,c,d^ Mean values with the same superscript letter are not significantly different from each other (*p* > 0.05 ANOVA followed by the Tukey test).

**Figure 4 antioxidants-12-00859-f004:**
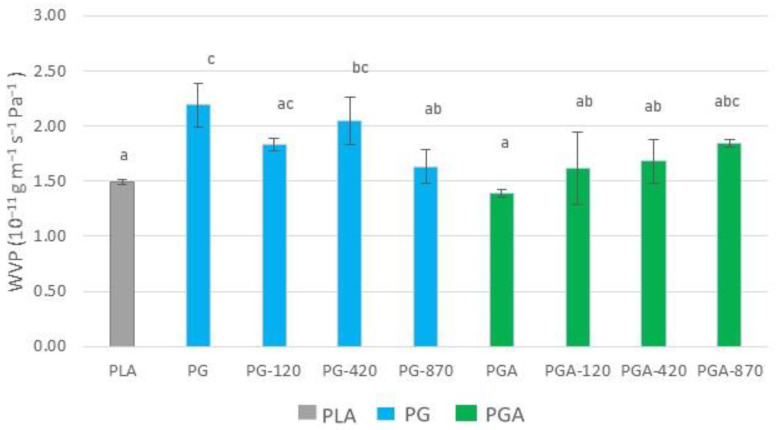
Measured WVP of PLA and coated PLA samples (gelatin side exposed to RH 30%, PLA side to RH 75%, (^a,b,c,d^ Mean values with the same superscript letter are not significantly different from each other (*p* > 0.05 ANOVA followed by the Tukey test).

**Figure 5 antioxidants-12-00859-f005:**
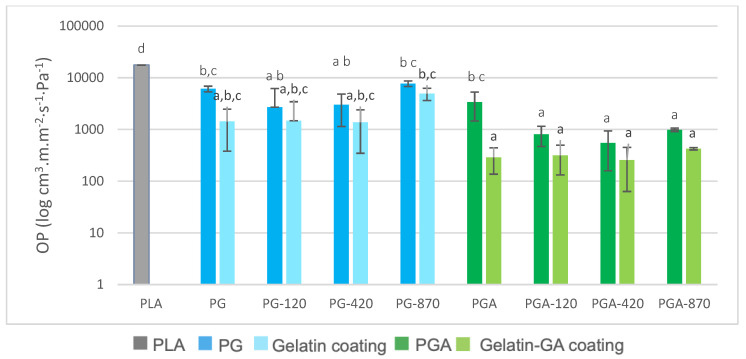
The oxygen permeability of PLA and PLA gelatin-coated without (PG) and with (PGA) gallic acid, and calculated OP of gelatin-based coating alone (^a,b,c,d^ Mean values with the same superscript letter are not significantly different from each other (*p* > 0.05 ANOVA followed by the Tukey test).

**Table 1 antioxidants-12-00859-t001:** Sample film codes and description.

Sample Code	Description
PLA	PLA base film (Nativia NTSS-25, double side corona treated during industrial manufacturing)
PG	PLA/gelatin (untreated)
PGA	PLA/gelatin-gallic acid (untreated)
PG-120, PG-420, PG-870	PLA/gelatin after 120 W, 420 W and 870 W corona treated
PGA-120, PGA-420, PGA-870	PLA/gelatin-gallic acid after 120 W, 420 W and 870 W corona treated

**Table 2 antioxidants-12-00859-t002:** Solubility in water (%), water absorption capacity (*m*_water_·*m*^−1^_dry coating_), and filtrate mass (g).

Coatings from Coated Films	Solubility	Water Absorption Capacity	Filtrate Mass
PG	39.13 +/− 13.69 ^b,c^	2470 +/− 527 ^b^	14.8 +/− 1.6 ^b^
PG-120	38.11 +/− 7.83 _b,c_	2399 +/− 143 ^b^	13.9 +/− 1.0 ^b^
PG-420	25.19 +/− 6.92 _a,b_	1546 +/− 215 ^a^	15.4 +/− 0.9 ^b^
PG-870	16.18 +/−9.37 _a,b_	1605 +/− 90 ^a^	14.0 +/− 0.7 ^b^
PGA	46.77 +/− 15.59 _b,c_	4837 +/− 671 ^c^	10.2 +/− 0.7 ^a^
PGA-120	30.80 +/− 1.93 _b_	4468 +/− 191 ^c^	9.9 +/− 1.6 ^a^
PGA-420	34.10 +/− 4.42 _b,c_	4720 +/− 505 ^c^	9.7 +/− 0.1 ^a^
PGA-870	34.64 +/− 18.84 _b,c_	4711 +/− 1506 ^c^	10.7 +/− 1.3 ^a^

Mean +/− std deviation. ^a,b,c^ Mean values with the same superscript letter are not significantly different from each other (*p* > 0.05 ANOVA followed by the Tukey test).

**Table 3 antioxidants-12-00859-t003:** Mean contact angle of water (°), surface free energy *γ*^S^ (mJ·m^−2^), critical surface tension *γ*^Crit^_,_ (mJ·m^−2^) with linear regression coefficients (*R*^2^), and surface water absorption rate (SWAR) of the PLA and coated films (10^4^ µL·mm^−2^·s^−1^).

Films	Water Contact Angle (°)	Polar/Dispersive Components Ratio	*γ*^S^ (*R*^2^) (mJ·m^−2^)	*γ*^Crit^ (*R*^2^) (mJ·m^−2^)	SWAR (10^4^ µL·mm^−2^·s^−1^)
PLA	76.7 ± 1.1 ^a^	0.34	32.04 (0.84)	27.34 (0.67)	11.06 ± 0.05 ^a^
PG	81.2 ± 7.6 ^b^	0.10	41.90 (0.62)	37.92 (0.66)	26.04 ± 3.25 ^c^
PG-120	84.1 ± 5.8 ^b^	0.08	39.54 (0.79)	40.50 (0.98)	14.56 ± 2.94 ^a,b^
PG-240	80.6 ± 2.2 ^b^	0.12	39.35 (0.97)	39.94 (0.98)	11.09 ± 0.28 ^a^
PG-870	87.8 ± 7.3 ^b^	0.03	43.00 (0.49)	42.53 (0.96)	11.27 ± 0.84 ^a^
PGA	81.5 ± 3.0 ^b^	0.09	40.79 (0.94)	41.03 (0.98)	12.62 ± 2.3 ^a^
PGA-120	76.0 ± 5.6 ^a,b^	0.21	37.89 (0.88)	38.04 (0.98)	11.17 ± 0.95 ^a^
PGA-240	79.0 ± 2.6 ^a,b^	0.16	38.22 (0.97)	38.28 (0.99)	10.4 ± 0.71 ^a^
PGA-870	75.6 ± 1.2 ^a^	0.18	40.07 (0.98)	39.95 (0.99)	10.86±1.02 ^a^

Mean +/− std deviation. ^a,b,c^ Mean values in a column with the same superscript letter are not significantly different from each other (*p* > 0.05 ANOVA followed by the Tukey test).

**Table 4 antioxidants-12-00859-t004:** The antioxidant activity (AA) of corona treated and non-treated PLA/gelatin and PLA/gelatin+gallic acid coated films, on coating alone after being delaminated from PLA, and on pure gelatin and pure gallic acid at same weight as in coatings, measured after 1 h and 24 h (DPPH concentration 50 mg·L^−1^; film area 24 cm^2^).

	Coated PLA Films		Coating Alone (after Delamination from PLA)	
Samples	AA (%) after 1 h	AA (%) after 24 h	AA (%) after 1 h	AA (%) after 24 h
Pure gelatin	/	/	30.83 ± 12.96 ^b^	37.29 ± 11.98 ^b^
Pure gallic acid	/	/	85.00 ± 1.48 ^d^	84.18 ± 0.98 ^d^
PG	0 * ± 0.28 ^a^	2.95 ± 0.06 ^a^	13.33 ± 3.33 ^a^	18.08 ± 8.53 ^a^
PG-120	/	/	13.89 ± 0.96 ^a^	32.77 ± 12.83 ^a,b^
PG-240	0 * ± 0.65 ^a^	3.20 ± 1.09 ^a^	9.44 ± 4.19 ^a^	22.60 ± 9.64 ^a^
PG-870	0.04 ± 1.94 ^a^	4.64 ± 4.49 ^a^	11.67 ± 7.07 ^a^	22.03 ± 9.59 ^a^
PGA	57.94 ± 24.36 ^b^	87.48 ± 0.67 ^b^	51.11 ± 6.74 ^c^	72.32 ± 5.18 ^c^
PGA-120	/	/	54.44 ± 0.96 ^c^	76.27 ± 1.69 ^c^
PGA-240	54.11 ± 21.07 ^b^	87.66 ± 0.45 ^b^	51.67 ± 4.41 ^c^	72.88 ± 2.94 ^c^
PGA-870	53.86 ± 28.89 ^b^	87.29 ± 0.40 ^b^	46.67 ± 4.42 ^c^	67.80 ± 1.69 ^c^

Mean +/− std deviation. * Values were negative or very close to zero (in the noise of the determination threshold), so considered as 0. ^a,b,c,d^ Mean values in a column with the same superscript letter are not significantly different from each other (*p* > 0.05 ANOVA followed by the Tukey test.

**Table 5 antioxidants-12-00859-t005:** The iron reducing (DO at 700 nm) and iron chelating (%) powers of corona treated and non-treated PLA/gelatin and PLA/gelatin and gallic acid coated films measured after 24 h.

Films	Reducing Power	Chelating Power (%)
Gelatin	0.78 ± 0.17 ^b^	51.51 ± 4.57 ^a^
Gallic acid	1.93 ± 0.04 ^d^	86.75 ± 1.71 ^d^
PG	0.43 ± 0.31 ^a^	52.48 ± 4.62 ^a^
PG-120	0.43 ± 0.06 ^a^	67.07 ± 3.00 ^b,c^
PG-240	0.43 ± 0.12 ^a^	55.29 ± 6.29 ^a^
PG-870	0.40 ± 0.07 ^a^	65.63 ± 3.00 ^b^
PGA	1.86 ± 0.05 ^c^	72.84 ± 0.83 ^c^
PGA-120	1.82 ± 0.11 ^c^	82.45 ± 6.30 ^c,d^
PGA-240	1.91 ± 0.02 ^c,d^	70.67 ± 4.91 ^b,c^
PGA-870	1.90 ± 0.03 ^c,d^	77.40 ± 2.91 ^b,c^

Mean +/− std deviation. ^a,b,c,d^ Mean values in a column with the same superscript letter are not significantly different from each other (*p* > 0.05 ANOVA followed by the Tukey test).

## Data Availability

The data are contained within the article and Appendix A.

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
