# Peer review of "Influence of Surface Corona Discharge Process on Functional and Antioxidant Properties of Bio-Active Coating Applied onto PLA Films"

_antioxidants, 2023, doi:10.3390/antiox12040859_

Round 1

Reviewer 1 Report

This paper demonstrates the effect of the surface corona discharge process on different functional and antioxidant properties of coated PLA film. The paper is showing new results with interesting properties like the surface barrier, permeability and antioxidative properties.

I have the following suggestions to improve the manuscript.

1. Though authors have mentioned biodegradability, they did not perform any test on that. Actually, as per the literature, PLA is biodegradable but needs tailored conditions like at elevated temperatures, not in normal temperatures. This should be noted in Intro when claiming biodegradability.

2. Since corona discharge is a key process in this study, please include a sketch of the application geometry used here to properly reflect the mechanism of application on the PLA substrate. Please also include the related parameters, such as electrodes, electric supply, frequency, etc relevant to the plasma application.

3. Some of the results need linking with the current literature, such as Section 3.3 and 3.4.1.

4. The methodology part is too long, some of the theories could be moved to supplementary and referred to in the text.

5. Though the contact angle result is shown in Table 3 with other results, the discussion was only related to SFT. It is important to discuss the data presented, otherwise remove it.

Author Response

see attached file, answers to the both reviewers

Reviewer 2 Report

I do not feel to be competent to judge about the article significance, originality and novelty. However, some knowledge is reported without proper references to its source, the investigated samples and conditions are not always adequately specified, equations are presented in a confused way. If other reviewers find the article to be sufficiently original and significant, the authors should rewrite it reader-friendly respecting all rules and recommendations for scientific papers before accepting for publication.

The article combines well written pieces with those suffering from lack of carefulness or written in a confused and confusing way.

In the introduction, the first paragraph should contain references to basic literature in addition to the reference to the recent paper. Reported well known facts are neither results of the authors nor facts known from prehistory. The fundamental papers or books from the field should be referred here.

Similarly, the knowledge reported in the first half of the second paragraph exceeds findings originating from reference 3. The authors should correctly report the source of referred knowledge, although well known.

Generally, knowledge reported in the description of the state of the art apparently originates from more sources than the authors acknowledge.

The conclusion at least in the first sentence repeats knowledge reported already in the introduction, I have found any results in the text specifically dealing with it, giving a reason for repeating in conclusions.

The authors do not respect the IUPAC and IUPAP recommendations on writing physical quantities and variables.
The IUPAP recommendation is available for example at https://iupap.org/wp-content/uploads/2021/03/A4.pdf.
The IUPAC recommended documents are Quantities, Units and Symbols in Physical Chemistry (https://iupac.org/what-we-do/books/greenbook/)
and On the use of italic and roman fonts for symbols in scientific text ( https://iupac.org/wp-content/uploads/2016/01/ICTNS-On-the-use-of-italic-and-roman-fonts-for-symbols-in-scientific-text.pdf)

The authors use symbols and abbreviations without proper explanation, as specified below. Abbreviations overuse is, in my opinion, not beneficial, it makes the text reader-unfriendly. The authors should search the nomenclature documents for the proper standard symbols instead of using their 2-4 letter abbreviations.
There is a lot of misprints in symbols writing; I point out at selected items only below, but the authors should correct all of them. The author should decide between /m2 and m-2 style and keep it through the whole article.

lines 101-103> The authors have no more information about the used PLA to be reported here?

line 104: Mw should be Mw in italics as a physical quantity. MV is not explained, the physical quantity should be in italics, the recommended symbol for molar volume is Vm.  Both -1 should be in superscript positions (-1).
lines 104-105: Density is reported without temperature to which it applies.
line 105: MP is not explained, the physical quantity should be in italics, the recommended symbol for melting temperature is Tm or Tmp.
           BP is not explained, the physical quantity should be in italics, the recommended symbol for boiling temperature is Tb or Tbp.
          LogP is not explained, in pKa, if K is an equilibrium constant, K should be in italics (maybe pKa) and the reaction to which it refers has to be indicated.
 The data specific or acquired for given batch of material (purity) should be referred separately from the general data of the substance obtained from tables and databases.
line 106: -1 should be in a superscript position (-1), otherwise the statement is not true.
lines 128-129, 131 in Table 1: spaces between values and units are missing. The same applies also to other passages of the text.

line 136: The referred paper [2] was not a paper first publishing the method or first using it for the described purpose. Ref. 2 should be preceded here by the proper references.

line 153: In Equation (1), the subscripts denoting phases have to be upright, and the upright cos function should stand instead of the product of undeclared co and s variables. I appreciate the proper reference given here.

line 156: 2 should be in superscript position for a square meter (m2). In my opinion, the non standardised SFE abbreviation should be avoided in the text - the authors should refer it either using commonly used symbol, or in full words. In the whole article, I found no place where using "SFE" would be beneficial.

lines 174-179: The whole passage is confused. The volumes should be always cursive as the physical quantities, eva subscripts should be always upright. The rates and other quantitites should be represented by one-letter cursive symbols taken of those recommended for such a quantity, and other specification like LW should be given in upright subscripts added to the basic symbol. The differentiation d operators should be upright.

lines 188-194: m should be here always cursive (m) as a physical quantity and the descriptions like ??????? ?????? ????????? as upright subscripts. The authors write both of them sometimes cursive and sometimes upright. I am not sure about writing style of physical quantities when they are reported in full words (Solubility), but I mean that the IUPAP and IUPAC recommendation applies to quantity symbols and that standard upright font is used when full words are used. Since subscripts meanings are explained in lines 189-190, I think that a choice of one-letter superscript denoting each of those states would be better in the equations than the full word subscripts.
The similar is valid for nearly all equations in the article, I do not list each of them specifically.

line 393: The passage is somehow confused.
...
line 504: The English used in 3.5 header is too advanced for me - I do not know the second word and it is not in the dictionary available.
Tables 2, 3, 4 and 5 and the figures contain letters as remarks, but I have not found their explanation. They look like taken from other paper, forgetting the legend.

Lines 571-575: Statements from the description of the state of the art are repeated, it does not look like the conclusion from this work.
Lines 576-578: It does not me seem to be the same what is stated in section 3.2
Lines 579-585: In Figure 2, I see higher value of the right column for PG-120 than for PG. It seems to me not to correspond with this conclusion.

Author Response

(The authors gave the same response as above.)

Round 2

Reviewer 2 Report

I appreciate responsible reflecting the comments concerning the factual content. Formal mistakes have been corrected only partially. I feel the choice of referred literature still not ideal; more primary sources of knowledge and books can be referred in addition to articles seeming to be too randomly selected that report facts from other literature. However, I do not insist on further improvement.

Explaining all symbols and abbreviations on the first use, even of those well known, should be checked - one example is given in specific comments.

I will quote from 1.3.1 of the IUPAC document Quantities, Units and Symbols in Physical Chemistry: "Subscripts and superscripts that are themselves symbols for physical quantities or for numbers should be printed in italic type; other subscripts and superscripts should be printed in Roman (upright) type."

Missing spaces between values and units have been added only somewhere.

The supplementary material consisting of only one composed picture would be more reader friendly when included into the main document - either into Materials and methods section, or after the end.

Line 116: reference to the page with products characteristics is appreciable, but it is probable that the article will be finding its readers for longer time than the URL remains valid. In addition, registration is required for data download. In my opinion, at least molar mass, density and surface tension/surface energy should be reported explicitly if they are available.

Line 118-119: Quantities have been correctly converted to italics. However, subscripts like mp, bp are descriptive ones and should be upright. Since even symbols used correctly in the accepted version are often converted to wrong during production, the correct usage should be set during proofreading.
Although standard symbols are now used, they should still be explained on the first use.

Line 142: Spaces between values and units missing
Line 146: Spaces between values and units missing
Line 148: Spaces between values and units missing

Line 155: References are better than previously, however, the literature about food packaging is in my opinion not the ideal one to be reported when introducing the method. I would expect ref. [26]  here and other references like those to Young , Laplace, Dupré, Adamson, Fox, Zisman, Neumann, Good, Stromberg  etc.

Line 173: In eq. (1), the cos function has been corrected. However, the subscripts denoting phases should be upright.
Line 175: θ is a variable, it should be cursive (θ).
Line 181: θ is a variable, it should be cursive (θ).
Line 199: d in the equations (2) and (3) denotes an operator, not a physical quantity, therefore it should be upright. eva subscript should be always upright (it seems to be sometimes upright and sometimes cursive). If A symbol has been chosen to denote areas, S and B as subscripts are more consistent with standards. On the other hand, if (t) denotes that it is a function of time, it is usual to be used in a base level, not as a subscript.
Line 202: t is a variable, it should be cursive (t)

Line 220: Mass standard symbol m should be used here, units an remain in the word explanation.
Line 358: Subscripts should be upright here.
Line 365: P is a variable, it should be cursive (P)
Line 386: I guess that the authors intend g as "gram". However, the standard symbol for mass is m , and when the mass in grams reported, then m [g] or m/g is usual.
   I understand now that a, b and c have been introduced to denote groups of values not significantly different each from other. If so, can be the description modified to make it quite clear?

I recommend to align columns in tables on ± sign or on decimal point of the basic value

Line 595, 596: Space between value and units missing
Line 622: Space between value and units missing
Line 647: Space between value and units missing (24h)
Line 677: Probably "video of" instead of "pf" should be here. In supplied materials, I see sketch and scheme, but no video.

The correct writing of variables and quantities can be solved during proofreading, there remain only several other points to be refined before an acceptation.
